# Recent Advances on the Role of B Vitamins in Cancer Prevention and Progression

**DOI:** 10.3390/ijms26051967

**Published:** 2025-02-25

**Authors:** Zachary Frost, Sandra Bakhit, Chelsea N. Amaefuna, Ryan V. Powers, Kota V. Ramana

**Affiliations:** Department of Biomedical Sciences, Noorda College of Osteopathic Medicine, Provo, UT 84606, USA

**Keywords:** B vitamins, cancer, antioxidants, metabolism, nutrients, Warberg effect

## Abstract

Water-soluble B vitamins, mainly obtained through dietary intake of fruits, vegetables, grains, and dairy products, act as co-factors in various biochemical processes, including DNA synthesis, repair, methylation, and energy metabolism. These vitamins include B1 (Thiamine), B2 (Riboflavin), B3 (Niacin), B5 (Pantothenic Acid), B6 (Pyridoxine), B7 (Biotin), B9 (Folate), and B12 (Cobalamin). Recent studies have shown that besides their fundamental physiological roles, B vitamins influence oncogenic metabolic pathways, including glycolysis (Warburg effect), mitochondrial function, and nucleotide biosynthesis. Although deficiencies in these vitamins are associated with several complications, emerging evidence suggests that excessive intake of specific B vitamins may also contribute to cancer progression and interfere with therapy due to impaired metabolic and genetic functions. This review discusses the tumor-suppressive and tumor-progressive roles of B vitamins in cancer. It also explores the recent evidence on a comprehensive understanding of the relationship between B vitamin metabolism and cancer progression and underscores the need for further research to determine the optimal balance of B vitamin intake for cancer prevention and therapy.

## 1. Introduction

B vitamins are a group of eight water-soluble nutrients essential in maintaining various metabolic processes in the body and are necessary for maintaining human health [1]. The B vitamins include B1 (Thiamine), B2 (Riboflavin), B3 (Niacin), B5 (Pantothenic Acid), B6 (Pyridoxine), B7 (Biotin), B9 (Folate), and B12 (Cobalamin). Each type of B vitamin participates in a specific biochemical reaction as a co-factor and helps with DNA synthesis, repair, methylation, and energy metabolism (Table 1). Since B vitamins play crucial roles in cellular functions and biochemical pathways, it’s no surprise that B vitamins are essential for human health and disease [1,2,3,4]. The deficiency of these vitamins could lead to multiple complications, including neurological disturbances, anemia, and skin diseases [3,4].

Further, certain drugs and diseases could cause a deficiency of these vitamins and increase complications. For example, prolonged metformin use has been shown to cause thiamine and cobalamin deficiency [5,6,7,8,9]. Similarly, corticosteroids could cause pyridoxine, folate, and cobalamin deficiency [10,11,12]. Moreover, several drugs, such as proton pump inhibitors [13,14], anticonvulsants [15,16], antibiotics [17], and chemotherapeutic drugs [18,19,20], have been shown to cause deficiency of specific B vitamins. B vitamins also play a role in cancer development and progression. In certain cancers, the deficiency of B vitamins was also observed.

The involvement of B vitamins in cancer is a complicated process. Generally, B vitamins play an important role in cancer prevention and progression by altering cellular metabolism and DNA synthesis, repair, and methylation [2,3]. For example, adequate folate, cobalamin, and pyridoxine levels support genomic stability by controlling the DNA strand breaks, mutations, and genomic instability. In addition, niacin contributes to NAD+ production, which is essential for DNA repair and energy balance. Although sufficient B vitamin intake could reduce the cancer progression and risk, excessive supplementation, specifically with vitamins B9, B12, and B6, has been associated with accelerated tumor growth in certain cancers, such as colorectal, prostate, and lung cancer. Further, B vitamins could also interact with oncogenic regulators such as HIF-1α, MYC, and AMPK, which influence cancer cell metabolism. Since B vitamins play a key role in normal cellular function, understanding their role in cancer is highly complex and requires a personalized approach to intake and supplementation, especially in cancer patients.

Further, recent studies also suggest that vitamin deficiencies and excessive intake of vitamins could be associated with cancer development and therapeutic resistance [21,22,23,24]. The deficiencies seen in B vitamins could disrupt processes such as DNA repair and immune regulation that cause genomic instability and an increased susceptibility to cancer. On the other hand, excessive supplementation of B vitamins has been associated with the risk of cancer progression [23,24]. This suggests that maintaining a balanced intake of vitamins and knowing the vitamin deficiencies are very important for maintaining a healthy lifestyle.

This review discusses the importance of water-soluble B vitamins in cancer prevention and progression. Further, we aimed to provide a comprehensive understanding of how these essential nutrients can influence cancer outcomes. We searched PubMed and Google Scholar to find articles published in the last 10 years or so, using keywords including various types of B vitamins, cancer, melanoma, leukemia, breast cancer, lung cancer, colon cancer, and other cancer types. We included various research articles, comprehensive narrative reviews, meta-analytical studies, systematic reviews, clinical studies, and pre-clinical studies to discuss their findings. We did not include studies on other fat-soluble and water-soluble vitamins, such as A and C. This narrative review article discusses only the role of B vitamins and their therapeutic significance in various cancers.

## 2. Vitamin B1: Thiamine

Vitamin B1 (thiamine) is a water-soluble vitamin in plant and animal-derived foods, such as meats, eggs, and dairy [25]. Thiamine maintains cellular viability by participating in critical metabolic pathways like glycolysis, the pentose phosphate pathway, and Kreb’s cycle. These pathways are essential for producing ATP and NADPH to sustain metabolic functions in the body [26]. Thiamine is particularly significant for aerobic metabolism, as it acts as a primary co-factor for pyruvate dehydrogenase in its active form, thiamine pyrophosphate (TPP). TPP also serves as a co-factor for alpha-keto dehydrogenase and transketolase, which convert ribulose-5-phosphate into RNA and DNA substrates and facilitate the conversion of amino acids and fatty acids into acetyl-CoA. Thus, thiamine directly contributes to ribose synthesis, a necessary substrate for cell proliferation, and provides metabolites for gluconeogenesis and NADPH synthesis [27,28].

The impact of thiamine deficiency can be profound, leading to severe conditions such as Beri-Beri disease and Wernicke-Korsakoff syndrome [29,30]. In cancer patients, thiamine deficiency has been strongly linked to the manifestation of delirium, highlighting its critical role in maintaining nervous system health [31]. Thiamine deficiency can also result in widespread neurological, cardiac, and gastrointestinal dysfunctions, which raises important questions about its role in cancer cell proliferation and progression [32,33,34,35].

A metabolic shift known as the Warburg effect (Figure 1) is often observed in the initial stages of cancer cell proliferation. This phenomenon, which has become a focus of alternative chemotherapeutic strategies, describes how cancer cells rely on glycolysis and fermentation of lactic acid for energy production, even in the oxygen presence [36,37]. This preference for glycolysis provides tumor cells with the metabolic intermediates necessary for rapid cell growth and survival. The role of TPP in the Warburg effect is shown in Figure 1. In addition, the individual roles of B vitamins on the Warburg effect are shown in Table 2. The overexpression of pyruvate dehydrogenase kinase (PDK) in cancer cells inactivates pyruvate dehydrogenase (PDH), effectively disconnecting glycolysis from the Krebs cycle and enhancing glycolytic activity [38]. The inhibition of PDK by TPP has shown promise in triggering apoptosis and reducing tumor cell proliferation by restoring PDH activity [39,40,41]. Through its role in maintaining PDH complex activity, thiamine has been shown to decrease tumor cell proliferation and glycolytic activity by countering PDK-induced PDH inactivation [42].

Further evidence of thiamine’s role in cancer metabolism comes from studies involving dichloroacetate, an analog of acetic acid known for its potential to increase aerobic metabolism. Although typically avoided due to its toxic side effects, dichloroacetate showed the action of reducing colorectal cancer cell growth by enhancing oxidative phosphorylation [43,44]. Thiamine’s involvement in cancer is also evident in studies where cancer cells maintained a base level of TPP despite thiamine-depleting vitamin stores in tissues. When keeping TPP levels low, cancer cells could produce the essential intermediates for growth. However, increasing TPP concentrations shifted metabolism towards PDH activation and mitochondrial respiration, stunting cancer cell proliferation [45,46]. Additionally, thiamine analog, oxythiamine has been shown to prevent cancer cell growth [47,48]. Further, it has been shown to reduce DNA and RNA synthesis by inhibiting ribose-5-phosphate and thereby reducing tumor cell growth [49]. Thiamine at high doses has been shown to prevent the growth of MCF-7 breast cancer cells [50]. In these cells, thiamine has also been shown to increase PDH activity and reduce glycolysis in the cancer cells. The expression of thiamine transporter SLC19A3 has been shown to be significantly expressed in chronic hypoxia-induced breast cancer cell lines (BT474) [51]. This increase in the transporter expression is correlated with the increased uptake of thiamine by the breast cancer cells. However, Liu et al. [52] have shown that SLC19A3 mRNA levels were downregulated in human breast cancer tumor tissues compared to normal tissues. They have suggested that decreased expression of this transporter could cause breast cancer cells resistance to apoptosis. Similarly, hypoxia has been shown to inhibit the uptake of microbiota-generated thiamine and TPP by the NCM460 human colon epithelial cells by reducing the expression of transporters such as SLC44A4, SLC19A2, and SLC19A3 [53]. Further, benfotiamine, a lipid-soluble derivative of thiamine, has been shown to control the proliferation of cancer cells and growth in a nude mice xenograft model [54].

Thiamine deficiency is most commonly seen in patients with diabetes [55,56], and diabetic patients are at risk of many complications. Further, some cancer patients have also been shown to have a deficiency of thiamine [57,58,59]. Therefore, diabetic patients with cancer have a significant risk of developing thiamine deficiency and energy metabolism, and supplementation of thiamine in such patients could prevent the cancer progression. Further, thiamine has been shown to regulate the expression of PKC, NF-κB, and advanced glycation end products (AGEs) in diabetics, and these signaling pathways are also critical in cancer progression [60,61]. Thiamine has been shown to prevent the activation of NF-κB and matrix metalloproteinases (MMPs) [62]. MMP activation is associated with tumor invasion and metastasis by modulating extracellular matrix remodeling [63,64]. Similarly, thiamine could also prevent the expression of prostaglandins and activation of COX-2 [65], which are involved in tumor growth. Similarly, thiamine has been shown to affect tumor growth by inhibiting the generation of reactive oxygen species (ROS) and reactive nitrogen species [66].

Thus, recent studies suggest that the impact of thiamine on cancer progression remains complex and variable (Figure 2). For instance, increased intake of thiamine has been linked with a significant risk of bladder cancer in men, whereas in women, it appears to have a protective effect [67]. This study also suggests that gender differences are influenced by dietary habits, where men typically consume more meat products leading to a risk of developing bladder cancer. On the other hand, women consuming more dairy products have a weaker association with bladder cancer. However, further research is needed to understand the gender-associated thiamine’s role in cancer. Thus, depending on the type of cancer and metabolic state, thiamine can have both tumor-promoting and tumor-suppressive effects. For example, thiamine supports tumor growth by enhancing glycolysis, the pentose phosphate pathway, and mitochondrial function and providing ATP, NADPH, and biosynthetic precursors essential for rapid tumor cell proliferation. However, in some conditions, thiamine deficiency could cause metabolic stress and oxidative DNA damage, potentially suppressing tumor growth. Although adequate dietary thiamine is necessary for normal function, excessive supplementation could promote tumor progression in susceptible cancers, and personalized approaches are needed when using thiamine in cancer therapy.

## 3. Vitamin B2: Riboflavin

Vitamin B2, or riboflavin, is primarily obtained from food sources like thiamine, with milk and dairy products among the most significant contributors [68,69]. Riboflavin is a coenzyme in various oxidation and reduction reactions in the body, mediating electron transfer and maintaining the integrity of several tissues, particularly neural tissues [70]. Like thiamine, riboflavin plays a critical role in cellular function by helping to keep glutathione in its reduced form, neutralizing reactive oxidative species, and protecting the body from free radical damage. Riboflavin also metabolizes other essential vitamins, including folate, vitamin B6, and B12 [71,72].

Riboflavin deficiency, known as ariboflavinosis, can lead to a marked increase in oxidative stress, with further adverse effects such as migraines, anemia, diabetes mellitus, hyperglycemia, and hypertension [70,71,72]. In severe cases, deficiency can result in growth retardation, anemia, renal damage, and degenerative effects on the nervous system [73]. While the extent of riboflavin’s influence on various cancer types varies, it is well-established that riboflavin is a crucial precursor to the coenzymes FAD and FMN, which regulate redox reactions in the TCA cycle and control levels of reactive oxygen species (ROS) in the body [70]. Riboflavin has been shown to regulate cellular ROS levels and increase the therapeutic efficacy of anti-cancer drugs [74,75,76]. A few studies also suggest that moderate riboflavin levels could trigger the extrinsic pathway of apoptosis and autophagy [77,78,79]. At the same time, higher doses could also activate specific apoptotic mechanisms by downregulating anti-apoptotic factors and promoting pro-apoptotic factors [80,81].

Further, the anti-inflammatory properties of riboflavin also help in cancer therapy. Notably, when combined with the chemotherapeutic drug cisplatin, riboflavin has been shown to mitigate cisplatin’s toxic effects by reducing the inflammatory response and replenishing antioxidant enzymes and cellular reductants [82,83]. Regulation of FAD/FMN levels, control of ROS production, and induction of apoptosis are major pathways where riboflavin is associated with increased cancer risk (Figure 3). Although some processes have been established, further research is needed to understand riboflavin’s dose-dependent and cancer-specific effects. However, few studies have indicated a correlation between riboflavin intake and cancer development [84,85,86]. Like thiamine, overconsumption and underconsumption of riboflavin can alter cancer risk, with the impact varying depending on the cancer type and affected region [86,87,88,89]. Riboflavin has been shown to prevent the growth of MCF-7 and MDA-MB-231 breast cancer cells, but not normal cells (L929) growth under the influence of visible light [90]. Similarly, Sturm et al. [91] have shown that riboflavin in combination with gemcitabine decreases the growth of bladder cancer cells growth in the presence of blue light at a wavelength of 453 nm. Another study by Chiu et al. [92] has shown that blue light and violet light illumination of riboflavin prevents the growth of colon cancer cells. In addition, radiated riboflavin prevents growth and increases apoptosis of C6 glioblastoma cells when compared to non-radiated riboflavin. Further, a few studies also indicate the role of riboflavin in prostate cancer. A recent study by Lv et al. [93] has shown a relationship between riboflavin intake and prostate-specific antigen (PSA) levels detected in American men. They have shown that there is an inverse relationship between riboflavin intake and PSA detection levels. Another study by Zhao et al. [94] has shown that riboflavin, by inhibiting the TGF-β signaling, could prevent pancreatic cancer metastasis. These studies suggest that a higher riboflavin intake may suggest a greater risk of diagnosing patients with advanced prostate cancer. Similarly, Gunathilake et al. [95], in a Korean population-based study, indicates that a higher riboflavin intake lowers the risk of developing colorectal cancer. Specifically, they have shown that males homozygous for the major alleles of MTRR rs1801394 and MTR rs1805087 polymorphisms and who had a higher intake of vitamin B2 had a significantly lower colorectal cancer risk. In a Chinese health study, Paragomy et al. [96] have shown that the increased riboflavin levels in the serum are correlated with the increased risk of pancreatic cancer. Similarly, Ma et al. [97] have also shown that increased riboflavin levels in the serum are associated with increased colorectal cancer.

As a precursor to FAD and FMN, riboflavin is linked to folate metabolism. Consequently, low riboflavin intake can exacerbate the effects of low dietary folate, which is necessary for DNA methylation and purines and pyrimidine synthesis. These factors are needed for cell growth and repair mechanisms [98,99]. Further, FAD also serves as a co-factor for the folate-metabolizing enzyme methylenetetrahydrofolate reductase (MTHFR). It catalyzes the one-carbon metabolism of 5,10-methylenetetrahydrofolate to 5-methyltetrahydrofolate, eventually converting homocysteine to methionine [98]. Few studies have also shown that riboflavin deficiency is associated with reduced MTHFR activity and elevated homocysteine levels, which could also serve as a risk factor for cancer development [100,101].

Further exploration of riboflavin’s role in cancer in the context of breast cancer has shown elevated riboflavin carrier protein (RCP) levels in patients with breast adenocarcinoma [102]. Since RCP is estrogen-induced, its elevated levels in breast cancer patients suggest that RCP could serve as a predictive marker for breast cancer [103] and may be involved in prostate cancer [104]. Additionally, riboflavin intake has been shown to decrease the risk of colorectal cancer in women, where deficiency disrupts MTHFR enzyme activity, contributing to cancer progression [86].

In esophageal cancer, riboflavin deficiency has been linked to changes in the esophageal epithelium, increasing the risk of cancer development [105,106]. Supplementation with riboflavin and niacin has shown effectiveness in reducing esophageal cancer incidence in regions with high rates of the disease [107]. Higher riboflavin intake, especially among female non-smokers, has also been associated with a reduced risk of lung cancer [108]. Riboflavin deficiency is also linked to cervical dysplasia, a precursor to cervical cancer [109]. In addition, riboflavin, combined with other B vitamins, such as thiamine, folate, or cobalamin, could offer protection against the progression of cancer [110,111,112]. Indeed, a recent study suggests that intake of riboflavin at a dose of 1.2 to 2.4 mg/day could contribute to a reduced risk of cervical cancer in Korean women [112]. Thus, recent studies suggest that riboflavin could be adjuvant therapy to increase the efficacy of anti-cancer drugs. However, its individual effects on cancer prevention and treatment need additional studies.

## 4. Vitamin B3: Niacin

Vitamin B3, known as niacin, is primarily derived from nicotinic acid, nicotinamide, and tryptophan. It is essential for generating coenzymes such as nicotinamide adenine dinucleotides (NAD and NADPH). These molecules are necessary for several metabolic pathways, including glycolysis, the Krebs Cycle, oxidative phosphorylation, and the hexose monophosphate (HMP) shunt [113]. The conversion of tryptophan to niacin, and subsequently to NAD, is catalyzed by a dioxygenase enzyme, which is upregulated by cortisol and tryptophan in the liver [114]. However, only a small amount of dietary tryptophan is typically converted to niacin, with most niacin obtained through other methods (Figure 4).

Further, niacin, through its precursor nicotinamide (NAM), contributes to the production of NAD+, which in turn acts as a substrate for key enzymes like sirtuin 1 (SIRT1) and poly ADP-ribose polymerase 1 (PARP1) [115]. These enzymes play a role in DNA repair, apoptosis, and carcinogenesis [116,117,118]. Thus, the therapies targeting NAD+ metabolism, such as PARP and NAMPT inhibitors, could enhance cancer cell death by blocking NAD+ dependency. Thus, the modulation of NAD+/NADH balance demonstrates a promising therapeutic strategy, but its effects depend on the specific cancer type and treatment approach. As with all essential nutrients, niacin deficiencies can significantly affect human health. The most severe effects of niacin deficiency are manifest in Pellagra, a disease characterized by the “three D’s”: dermatitis (often presenting as a casal necklace), diarrhea, and dementia [119]. Pellagra can also occur in Hartnup disease, where the failure to absorb amino acids like tryptophan through the intestine and kidneys leads to a secondary niacin deficiency [120].

Nikas et al. [121] have shown that PARP1 is activated by DNA strand breaks, initiates DNA repair mechanisms, and maintains genomic stability. However, excessive DNA damage can over-activate PARP1, leading to the depletion of NAD+ and other metabolic factors, which may result in necrosis. NAM is notable for inhibiting SIRT1 and PARP1 through negative feedback mechanisms, thus preventing over-regulated replication processes [121,122].

NAM has been shown to prevent carcinogenic events in melanoma [123,124]. For example, NAM has been shown to prevent UV-radiation-induced damage to the keratinocytes [125]. It promotes DNA repair and cellular energy production and decreases inflammation and cell death. In a randomized phase-III clinical trial, Chen et al. suggest that oral NAM reduces the risk of developing new nonmelanoma skin cancers and actinic keratoses in high-risk patients [126]. Similarly, Carneiro et al. [127] have also suggested that NAM is preventive to the development of non-melanoma skin cancers in cancer patients. However, a meta-analytical systemic study by Tosti et al. [128] suggests no significant chemopreventive effect of NAM supplementation in skin cancers. Along similar lines, a comprehensive epidemiological study indicates that niacin supplementation provided partial benefit from developing SCC but had no effect on BCC and melanoma [124]. However, a recent study indicates that NAM prevents melanoma cell growth in culture as well as in mice models [123].

Recent studies also suggest that niacin deficiency regulates DNA damage repair caused by nicotine-derived nitrosamine ketones by modulating the genes involved in the cancer progression [129]. This study indicates that niacin deficiency could be a risk for cigarette smoke-induced genomic instability and cancer development. Moreover, Nicotinamide phosphoribosyltransferase (NAMPT) is a key enzyme involved in the production of NAD in the salvage pathway required for cancer cell growth and progression [130] (Figure 3). Overexpression of NAMPT has been shown in many cancer types and inhibitors of NAMPT have been shown to prevent cancer growth [131]. Cole et al. have also indicated a possible treatment option of coadministration of NAMPT along with niacin for small-cell lung cancer and neuronal cancers [132]. Similarly, Nomura et al. [133] have shown that reducing the levels of nicotinic acid riboside could enhance the efficacy of NAMPT inhibitors in treating neuroendocrine cancers.

In addition, Tabrizi and Abyar [134] have shown that copper-based compounds containing vitamins B3 and B4 prevent the growth of breast cancer cells such as MCF7 and MDA-MB-231. Comparably, Abdel-Mohsen et al. [135] have also suggested that the copper(I) nicotinate complex, by regulating Notch1 signaling, could prevent triple-negative breast cancer cell growth. NAM has been shown to prevent triple-breast cancer cells by reducing the mitochondrial membrane potential and ATP production and increasing reverse electron transport, lipid metabolism, and reactive oxygen species [136]. Interestingly, a curcumin derivative containing two molecules of niacin has been shown to prevent the growth of colon, breast, and nasopharyngeal cancer cells by promoting P53-mediated apoptosis and cell cycle arrest [137]. In addition, Kim et al. [138] have also shown that niacin prevents TRAIL-induced apoptotic cell death by activating the autophagy flux in human colon cancer cells.

Similarly, niacin has been shown to prevent glioblastoma in pre-clinical studies. Sarkar et al. [139] have shown that niacin-stimulated monocytes produced by interferon-alpha14 inhibit the growth of brain tumor-initiating cells. In the same study, authors have found that in a mouse model of glioblastoma, niacin prevents tumor growth. NAM has been shown to prevent melanoma growth in culture as well as in mouse models, maybe by inhibiting the activation of SIRT2 [123,140]. Selvanesan et al. [141] have also shown in a mouse model that a combination of gemcitabine with nicotinamide (NAM) prevents pancreatic cancer by decreasing the tumor-associated macrophages and myeloid-derived suppressor cells. Similarly, Shu et al. [142] have shown that the niacin-ligated platinum (iv) and ruthenium (ii) chimeric complex prevents metastasis as well as the growth of cancer cells in vivo.

Accordingly, current evidence suggests that niacin could be a potent chemopreventive agent in controlling the growth of various cancer types [124,143]. Further, niacin, along with chemotherapeutic drugs, has been shown to increase overall survival and efficacy. Indeed, a recent National Health and Nutrition Examination Survey from 1999–2014 conducted by Ying et al. [144] has suggested that higher dietary vitamin B3 intake is associated with an improved survival rate and lowered mortality in cancer patients.

## 5. Vitamin B5: Pantothenic Acid

Pantothenic acid (vitamin B5 or pantothenate) in its anionic form, is a precursor to coenzyme A (CoA) and is an essential micronutrient found generally in a wide variety of foods, including animal products, mushrooms, potatoes, and oats [145]. A portion of pantothenic acid is also synthesized de novo by the gut microbiota after a three-step enzymatic reaction, which concludes with an adenosine triphosphate (ATP)-dependent condensation of β-alanine and pantoate by pantothenate synthetase [146]. Along with cysteine and four ATP molecules, pantothenic acid is necessary for the biosynthesis of CoA in a five-step enzymatic reaction [146]. CoA is then utilized as a fundamental coenzyme in numerous metabolic processes, most notably in the pyruvate oxidation in the Krebs cycle for energy production and during both the synthesis and oxidation of fatty acids.

Despite its essential role in metabolism, pantothenic acid could have multiple effects on cancer progression and prevention. Few studies have shown its cancer-promoting actions. For example, a recent study by Heckmann et al. [147] has demonstrated that pantothenic acid increases the levels of CoA in fibroblasts. [147]. Murine models, including TLX-5 lymphoma, sarcoma 180, and fibrosarcoma, also demonstrated lower levels of pantothenate, CoA, and acetyl-CoA compared to healthy mice. Miallot et al. [148] have indicated that pantothenic acid increases the polarization of myeloid and dendritic cells and antigen presentation in sarcoma. Interestingly, these tumor models showed an increase in 4-phosphopantothenate levels, an intermediate in CoA synthesis catalyzed by pantothenate kinase. This abnormal level may result from CoA degradation or increased pantothenate kinase activity rather than an increase in CoA itself [149].

However, some human studies have shown no significant association between a high pantothenic acid-containing diet and the risk of esophageal squamous cell carcinoma [150]. Similarly, studies on urothelial cell carcinoma and gastric cancer revealed no statistically significant correlation between pantothenic acid intake and cancer risk [151,152]. In breast cancer, a cohort study involving 27,853 women aged 45 years or older, with 462 cases of diagnosed breast cancer, found no association between pantothenic acid intake over 12 months and breast cancer development [153]. Furthermore, in a study involving five patients with advanced carcinomas (three with stage 4 cervical carcinoma and two with recurrent carcinoma of the floor of the mouth), a liquid diet deficient in pantothenic acid for 4 to 10 weeks did not alter the tumor growth rate compared to periods when the diet included pantothenic acid. Although urinalysis confirmed a decrease in pantothenic acid excretion, suggesting lower body levels, no symptoms of pantothenic acid deficiency were observed, likely due to endogenous production from intestinal microbiota [154].

Further, tumor-promoting effects of pantothenic acid have been shown in certain cancer cell types. For example, in breast cancer, increased pantothenic acid concentrations correlated with enhanced glycolytic activity and cell migration, particularly in the MCF-7 luminal gene cluster cell line, the JIMT-1 basal A cell line, and the MDA-MB-231, MDA-MB-435, and MDA-MB-436 basal B cell lines [155]. Similarly, Kreuzaler et al. [156] have shown that pantothenic acid is involved in the upregulation of c-MYC and SLC5A6, which promote cancer growth. Further, dietary restriction of pantothenic acid prevents tumor growth by reversing the c-MYC-modulated metabolic changes. Similarly, pantothenic acid and its metabolites have demonstrated protective qualities in specific cancer cells. For example, in Ehrlich ascites tumor cells grown in the peritoneal cavity of Swiss female mice, pantothenic acid and its derivatives (pantothenol and pantethine) decreased plasma membrane and mitochondrial outer membrane permeability. This protective effect is thought to result from the biosynthetic increase in cholesterol, which counters the leakage induced by digitonin, a compound that disrupts membrane integrity by complexing with cholesterol [157]. Additionally, pantothenic acid and its derivatives protected Ehrlich ascites tumor cells against lipid peroxidation induced by free oxygen radicals, further supporting its role in promoting cellular repair mechanisms [158].

Elevated levels of pantothenic acid levels were also found in the MKN-28 and AGS cell lines of gastric cancer cells, suggesting that increased phospholipid production from reactive oxygen species could contribute to both the rise in pantothenic acid levels and the proliferation of these cells [159]. A case-control study by Secchi et al. [160] in Cordoba, Argentina, revealed that high dietary intake of pantothenic acid was statistically significant in increasing the risk of oral squamous cell carcinoma.

Pantothenic acid and its metabolites have also been implicated in treating various cancers in some clinical studies. For example, higher pre-treatment plasma levels of pantothenic acid in patients with Stage III or IV melanoma correlated with an increased response to anti-programmed cell death protein 1 (PD1) antibody therapy. Similarly, pre-treatment with pantothenate injections enhanced the efficacy of anti-PD1 antibody therapy in the murine colon carcinoma model MC38 [161]. In cancers that rely heavily on glycolysis and lactic acid fermentation for growth, such as soft tissue sarcoma (STS), targeting pantetheine, a degradative product of CoA, has shown promise in reducing tumor growth. Treatment with Vnn1 pantetheinase, which degrades pantetheine into pantothenate and cysteamine, led to increased mitochondrial CoA levels and a move from glycolysis to oxidative phosphorylation, thereby suppressing tumor growth (Figure 5) [162]. Pantethine, a disulfide bridge-linked molecule of two pantetheine molecules, has also shown potential in treating multidrug-resistant cancers by inhibiting the release of microparticles that protect cancer cells from treatment [163]. In mouse models of ovarian tumors derived from human ovarian tumor xenografts, pantetheine treatment decreased metastasis and ascites formation [164].

## 6. Vitamin B6: Pyridoxine

Vitamin B6 is an essential water-soluble vitamin that plays a role in over 140 different biochemical reactions within the body. These reactions include the synthesis of heme precursor δ-aminolevulinic acid, sphingoid bases essential for myelin production, and several neurotransmitters such as serotonin, norepinephrine, epinephrine, and γ-aminobutyrate (GABA) [165]. Vitamin B6 is also a cofactor for several enzymatic reactions involved in amino acid metabolism, glycolysis, gluconeogenesis, glycogenesis, trans-sulfuration, immune response, and polyamine biosynthesis [166]. Structurally, vitamin B6 encompasses six chemically similar compounds, all containing a pyridine ring with varying groups at the 4′ position, and these compounds are interconvertible. The most bioactive form is pyridoxal 5′-phosphate (PLP), with other forms including pyridoxal, pyridoxamine, pyridoxamine 5′-phosphate, pyridoxine, and pyridoxine 5′-phosphate [167]. Failure to consume adequate amounts of vitamin B6, as well as conditions such as alcoholism and renal or liver complications, can lead to various health issues, including sideroblastic anemia, seizures, peripheral neuropathy, mood changes, and seborrheic dermatitis [165].

Further, pyridoxine also plays an important role in amino acid metabolism. It acts as a coenzyme in transamination, decarboxylation, and one-carbon metabolism, which supports nucleotide biosynthesis and cellular redox balance [166]. Further, PLP is crucial for protein synthesis, neurotransmitter production, and glutathione-mediated antioxidant defense. Pyridoxine influences cell proliferation in cancer cells by increasing amino acid metabolism and nucleotide synthesis. It also modulates apoptosis by regulating oxidative stress, redox balance, and epigenetic alterations. Thus, the deficiency of pyridoxine could promote DNA damage and tumorigenesis, while an excess of pyridoxine could promote tumor growth by modulating the redox-mediated metabolic pathways. Therefore, pyridoxine seems to play a major role in cancer progression as solid tumors rely on high amino acid turnover for survival and growth.

On the other hand, vitamin B6 has also been linked to a reduced risk of developing and proliferating diverse types of cancer. A meta-analysis study that monitored the dietary intake of vitamin B6 across 1,959,417 individuals, including 98,975 cancer cases, revealed that high intake of vitamin B6 was associated with a reduced risk of cancer at multiple sites, including the breast, colorectal area, ovary, prostate, immune system, endometrium, lung, stomach, esophagus, pancreas, kidney, bladder, oral cavity, nasopharynx, larynx, cervix, liver, and brain [168]. Specifically concerning breast cancer, a case-control study nested in the Multiethnic Cohort in Hawaii and Southern California indicates that increased circulating levels of PLP are associated with a reduced risk of invasive postmenopausal breast cancer [169]. Another cohort study suggested that vitamin B6 intake over 12 months had a protective effect against breast cancer development [153]. Additionally, a meta-analysis of eighteen studies (11 prospective studies and seven case-control studies) on the association of dietary vitamin B6 intake with pancreatic cancer showed that high blood levels of pyridoxal 5′-phosphate might protect against the development of pancreatic cancer [170]. While vitamin B6 supplementation has shown inconsistent and mostly nonsignificant associations with colorectal cancer development, high blood levels of pyridoxal 5′-phosphate have consistently demonstrated a 30–50% reduction in colorectal cancer risk [171]. Another meta-analysis found variability in the association between dietary vitamin B6 intake and colorectal cancer risk across nine studies but confirmed an inverse relationship between blood pyridoxal 5′-phosphate levels and colorectal cancer risk across four studies [172]. A recent meta-analysis study also indicates the risk of developing colon cancer is negatively correlated with the vitamin B6 and PLP levels [173]. Interestingly, Holowatyj et al. [174] have indicated that high preoperative vitamin B6 is linked to increased survival in stage 1-III colorectal cancer patients. Similarly, Li et al. [175] have also indicated that high PLP levels but not Par could be associated with the increased survival of colon cancer patients. Along similar lines, Xu et al. [176] have also suggested that increased PLP and pyridoxal are associated with colorectal cancer risk in the Chinese population. Thus, the protective effects of pyridoxal 5′-phosphate are thought to be due to its involvement in one-carbon metabolism, which is related to DNA synthesis and methylation, and its role in reducing inflammation, proliferation, and oxidative stress [171]. Indeed, Wu et al. [177] have indicated that Vitamin B6 is a cofactor associated with enzymes, like serine hydroxymethyltransferase (SHMT), methionine synthase reductase (MTRR), and methionine synthase (MS) and that it also plays a regulatory role in regulating genomic stability and cell viability in breast cancer patients.

The ability of vitamin B6 to interact with nuclear receptors and express antioxidant, pro-apoptotic, and anti-angiogenic effects in cells has been further elucidated. Vitamin B6 conjugates with RIP140, a receptor-interacting protein, enhancing its transcriptional co-repressive activity by increasing its interaction with histone deacetylases and retaining RIP140 in the nucleus [178].

Further, vitamin B6 deficiency has been linked to an increased risk of various cancers. For example, a recent study indicates that PLP deficiency leads to malignant tumors in a Drosophila model [179]. Similarly, Yasuda et al. [180] have shown that vitamin B6 deficiency is common in patients with primary and secondary myelofibrosis. Moreover, vitamin B6 deficiency has been linked to sarcopenia, a condition in which muscle loss is due to factors such as aging or immobility [181]. Spinneker et al. [182] have suggested an increase in colon tumorigenesis in vitamin B6-deficient mice, while vitamin B6 deficiency in rats showed signs of chronic pancreatitis, a condition known to increase the risk of pancreatic cancer. Several mechanisms have been proposed to explain the correlation between vitamin B6 deficiency and carcinogenesis. In DNA synthesis, vitamin B6 deficiency reduces serine hydroxy-methyltransferase activity, leading to a lack of methylene groups for 5,10-methylenetetrahydrofolate. This deficiency causes inadequate methylation of deoxy-uridylate to deoxy-thymidylate, resulting in the misincorporation of uracil into DNA instead of thymidine, which leads to chromosome strand breaks and impaired DNA excision repair. Another proposed mechanism involves DNA hypomethylation, given that vitamin B6 is associated with DNA methylation. A third mechanism suggests that vitamin B6 deficiency impairs the activity of detoxifying enzymes such as glutathione S-transferase and glutathione peroxidase, which detoxify carcinogenic compounds. Pyridoxal 5′-phosphate is required for enzymes in the trans-sulfuration pathway to generate cysteine, which is essential for glutathione synthesis (Figure 6). A fourth mechanism proposes that vitamin B6 deficiency increases steroid hormone sensitivity, potentially contributing to the development of breast, uterine, and prostate tumors [182].

Increased vitamin B6 catabolism, marked by a rise in the PAr index (the ratio of 4-pyridoxic acid, a catabolic product, to the sum of pyridoxal and pyridoxal 5′-phosphate), has been associated with a higher risk of lung cancer. A study of 20 cohorts across four continents found that an increase in the PAr index could serve as a pre-diagnostic marker for lung cancer [183]. Additionally, a survey of 5364 matched case-control pairs showed that impaired vitamin B6 status was significantly associated with an increased risk of lung cancer, with squamous cell carcinoma being the most common form [184]. In patients with non-small cell lung cancer, treatment with cytotoxic agents such as cisplatin was less effective if the patient had low expression of pyridoxal kinase, which converts pyridoxal to its more active form, pyridoxal 5′-phosphate [185]. Huang et al. [186] have also shown that low serum B6 vitamers are linked with increased pancreatic cancer risk in the Asian population. Similarly, lower levels of PLP have been associated with the development of pancreatic ductal adenocarcinoma [187]. These studies suggest that high levels of vitamin B6 could provide protection against various cancers, while low levels or deficiency are a major risk factor for promoting cancer growth.

## 7. Vitamin B7: Biotin

Vitamin B7, known as biotin, is water-soluble and typically obtained through dietary sources such as beef, eggs, salmon, pork chops, and vegetables like broccoli and spinach [188]. It acts as a cofactor for various carboxylase enzymes involved in metabolic pathways, including fatty acid synthesis, glucose metabolism, and amino acid metabolism [189]. Additionally, biotin plays a role in epigenetics and influences gene silencing, DNA repair, and chromatin remodeling [189]. The importance of biotin is further highlighted by the substantial energy required to produce a single molecule for participation in these biochemical reactions [190].

While biotin is a vital cofactor in many biochemical reactions, its broader roles in other pathways are less understood than other B vitamins. However, evidence suggests that biotin may be necessary for skin maintenance. Ogawa et al. [191] observed that acrodermatitis enteropathica (AE), partly due to biotin deficiency, led to symptoms similar to those in zinc-deficient diets. Further, biotin deficiency results in more pronounced irritant contact dermatitis, mirroring the effects of zinc deficiency. Therefore, biotin may play a key role in regulating and maintaining skin health, especially in individuals with skin conditions that impact their quality of life [191,192].

Biotin has also been associated with effects on glycolytic and gluconeogenic enzyme activity. Sugita et al. [193] have shown that in biotin treatment in streptozotocin-induced rats, the mRNA levels of phosphoenolpyruvate carboxykinase and glucose-6-phosphatase were decreased, and glucokinase was increased. Similarly, McCarty [194] has suggested that a high dose of biotin also alters the expression of glycolytic enzymes. In studies on chronic kidney disease (CKD), the regulation of cGMP, particularly when complexed with specific protein kinases, showed promise for reno-protective functions, potentially aiding in the prevention of CKD [195]. Moreover, inhibitors of the cGMP pathway, especially when part of the cGMP-cGK1-PDE complex, have shown potential for anti-clotting and anticancer properties. These findings could help to develop new cancer treatment modalities. However, further studies are needed to demonstrate biotin’s role in cGMP activity, signaling, coagulation, and thrombotic events, as well as its potential contributions to cancer cell proliferation.

Indeed, Maiti and Paira [196] have indicated that sodium-dependent multivitamin transporter (a biotin transporter) is overexpressed in various cancer cells, including breast, colon, lung, ovarian, and leukemia. Thus, biotin is used in the development of various drug delivery methods to treat cancers. For example, Tang et al. [197] have developed biotin-modified liposomes for breast cancer. Raza et al. [198] have indicated that the seleno-biotin compound induces apoptosis in ovarian cancer cells. Similarly, Kundu et al. [199] have shown the potential use of Chitosan-biotin-conjugated nanoparticles for increasing the chemotherapy potential.

Biotin is well-tolerated, with few noted adverse effects, toxicities, or contraindications. However, biotin deficiency can arise from insufficient dietary intake or an inability to utilize the vitamin within the body. Individuals deficient in biotin typically present symptoms such as skin rashes, hair loss, and fragile nails, along with neurological deficits like depression and paresthesias [188]. Recent studies mostly concentrated on developing drug delivery methods using biotin conjugates to prevent cancer. However, the connection between biotin deficiency and cancer is not clear, and additional studies are needed to understand the significance of biotin in cancer development and therapy.

## 8. Vitamin B9: Folate

Vitamin B9, known as folate, is an essential water-soluble vitamin found in legumes and green leafy vegetables. Folate deficiency is associated with developmental and degenerative conditions such as neural tube defects in embryos [200]. Consequently, vitamin B9 supplementation is recommended for women of childbearing age, starting one month prior to conception to the first trimester of pregnancy. Folate is involved in one-carbon metabolism and has been studied for its role in cancer development [201,202]. It is required in the methionine pathway to convert homocysteine to methionine, which is converted to S-adenosylmethionine (SAM). SAM plays a fundamental role in DNA and RNA methylation, affecting gene transcription and the expression of tumor suppressors and proto-oncogenes (Figure 6). Additionally, folate is vital in converting deoxyuridine monophosphate (dUMP) to deoxythymidine monophosphate (dTMP), which is essential for DNA synthesis and repair. Therefore, folate deficiency can lead to uracil misincorporation instead of thymine, resulting in DNA repair dysfunction, unstable DNA, and strand breaks [203,204]. On the other hand, excessive folate can also cause tumor progression by supporting rapid cell proliferation. Thus, deficiency and oversupply of folate can influence tumor development and progression.

Epigenetic changes caused by chronic inflammation can also be a factor in cancer development. Further patients with inflammatory bowel diseases are at a higher risk of developing colorectal cancer. Tumors of the gastrointestinal tract, prostate, and liver have also been linked to sites of chronic inflammation. B vitamins, including folate, regulate inflammation and the immune response due to their involvement in nucleic acid, protein synthesis, and methylation. Disruptions in the immune system can occur due to improper cytokine production, faulty antigen presentation, and an unregulated immune response. Inefficient methylation, often caused by vitamin B deficiencies, can lead to hyperhomocysteinemia, a condition associated with oxidative stress and chronic inflammation [205]. Elevated homocysteine levels and low folate have shown an increased risk of lung, breast, and colorectal cancers [206,207]. Further, elevated homocysteine and methionine have been suggested to disrupt the epigenetic modification of specific genes that regulate breast cancer progression and initiation, such as RASS-F1 and BRACA1 [208,209].

Moreover, some case-control studies have shown an inverse correlation between dietary folate intake and the risk of pharyngeal, oral, esophageal, colorectal, pancreatic, laryngeal, and breast cancers [210,211,212,213,214,215]. Additionally, risk estimates for cancers of the endometrium, ovary, prostate, and kidney were below unity, indicating a potential protective effect, while stomach cancer showed no relation to dietary folate intake [216,217]. Studies using colon cancer cells have demonstrated that folic acid supplementation can inhibit cell proliferation. This inhibition occurs through activating the c-SRC-mediated pathway and amplified levels of the cyclin-dependent kinase (CDK) inhibitor and tumor suppressor p53, leading to G0/G1 cell cycle arrest [218]. Folate has been shown to be critical in regulating the cancer growth [210,211,212,213,214,215,219,220,221,222]. Specifically, colorectal cancer has been one of the most extensively studied cancers concerning folate’s role in carcinogenesis. Most studies have shown a significant inverse relationship between folate status and colorectal cancer risk [220,221,222]. Epidemiologic studies comparing subjects with the highest dietary folate intake to those with the lowest have suggested a reduction of about 40% in the risk of colorectal neoplasms [223]. Few studies also suggest that even a modest decrease in folate levels can enhance colorectal cancer risk without clinical evidence of folate deficiency [224,225]. Randomized intervention studies in humans have also shown that folate supplementation in patients with resected colonic adenomas can reverse colorectal cancer biomarkers, including genomic DNA hypomethylation in the rectal mucosa and decreased proliferation of rectal mucosal cells [226,227].

Although these studies provide evidence of an inverse association between folate levels and colon cancer risk, they are still insufficient to draw definitive conclusions. However, a safe and effective dosage for folate has not been established in humans. Some studies have suggested that excessive folate supplementation could increase cancer risk and promote tumor progression [219]. Folate supplementation is available in various forms, including tablets, mixtures, and intravenous preparations. Recently, several countries have implemented food fortification programs in grains and cereals to prevent neural tube defects in embryos [228].

Further, few studies suggest that the protective effects of folate plateau are beyond a certain level and do not increase indefinitely with higher intake [219,222]. Colorectal cancer studies observed a decreased cancer risk with moderate folate status but an increased risk with excessive intake [223]. Similar findings were observed in breast cancer, where high plasma folate concentrations were associated with premenopausal breast cancer. A double-blind study on colorectal polyp recurrence showed that individuals supplemented with 1 mg/day of folate had more multiple and advanced adenomas than the placebo group [229]. These results suggest that people with prior adenomas might be at a higher risk of developing various or advanced polyps due to folic acid supplementation. The study also indicated a higher rate of invasive prostate cancer among the folic acid group [229]. Multiple in vitro and animal studies have summarized that increased folate supplementation may promote the progression of established tumors, while folate deficiency might contribute to tumorigenesis [230].

## 9. Vitamin B12: Cobalamin

Vitamin B12 (Cobalamin) is found in animal-derived products such as dairy, red meat, and eggs [231]. It is not produced by plants, making vegans and vegetarians at a higher risk for B12 deficiency unless they rely on supplementation, as they depend solely on microbial sources. Like folate, vitamin B12 is necessary due to its role in one-carbon metabolism (Figure 6), which involves homocysteine, methionine, and B vitamins [232]. One-carbon metabolism directs the one-carbon units toward the folate cycle, essential for DNA and RNA synthesis, or toward methionine regeneration, where methionine acts as a precursor to S-adenosylmethionine (SAM). SAM is vital for the methylation of histones, proteins, and DNA. Homocysteine generates methionine through a reaction catalyzed by the enzyme methionine synthase (MeS). The activation of MeS requires cobalamin as a cofactor for converting 5-methyl tetrahydrofolate (THF) to THF. Further, SAM is a potential methyl group donor for DNA, RNA, and protein methylations. Thus, SAM-mediated methylation is also significant for gene expression regulation, genomic stability, and epigenetic modifications, which influence cancer progression. Cobalamin is also vital for nucleotide synthesis, which is required for DNA replication and repair. Thus, a deficiency of cobalamin could lead to DNA damage, uracil misincorporation, and hypomethylation, which increases genomic instability and promotes the risk of developing cancer-causing mutations. Similarly, an excess of B12 could promote tumorigenesis by increasing cell proliferation and modulating cancer cell metabolism.

Further, both vitamin B12 and folate deficiencies are associated with a reduction in a significant antioxidant protein called glutathione, a product of the trans-sulfuration pathway. Glutathione acts as an inhibitor of reactive oxygen species (ROS)-induced oxidative stress and a mediator of redox balance. The impact of folate and vitamin B12 was examined in a study in which mice were treated with Azoxymethane (AOM), a compound known to induce tumors and oxidative stress. The study showed that both vitamins reduced the cytotoxic effects of AOM and mitigated oxidative stress [233]. A recent study by Obeid et al. [234] has indicated high plasma levels of vitamin B12 in various cancer patients. However, the role of B12 in promoting or preventing cancer is still not clear.

Moreover, a few studies also indicated a significant role of vitamin B12 intake and risk of developing colorectal, breast, prostate, malignant melanoma, or squamous cell carcinoma [235,236,237,238]. However, the study did reveal a positive correlation between vitamin B12 and the risk of esophageal cancer [239]. Another nested case-control study examined the association between serum concentrations of one-carbon nutrients, including vitamin B12, and upper gastrointestinal cancers. A statistically significant correlation was noticed between low vitamin B12 serum concentration and an increased risk of non-cardia gastric adenocarcinoma (NCGA). This association is mechanistically understandable due to the growth of the intestinal type of gastric cancer model. Chronic superficial gastritis often progresses to chronic atrophic gastritis and eventually to adenocarcinoma. Chronic atrophic gastritis leads to decreased gastric acid secretion, which diminishes vitamin B12 absorption. This study suggested that vitamin B12 deficiency in these cases was due to malabsorption rather than insufficient dietary intake [240].

While vitamin B12 deficiency can contribute to tumor development, over-supplementation may also increase cancer risk. For example, Collin et al. [241] have shown that increased intake of folate and vitamin B12 increases the risk of prostate cancer. Similarly, use of vitamin B12 supplements, excluding those from multivitamins, has been associated with a 30–40% increase in lung cancer risk. In contrast, no association was found between folic acid supplementation and cancer risk [242,243]. Similarly, a Norwegian randomized controlled trial has found an enhanced risk of lung cancer development in individuals who have taken both vitamin B12 and B9 supplements [244]. Additionally, Fanidi et al. [245] have also suggested increased lung cancer risk with high supplementation of vitamins B12 and B6. To further confirm the role of vitamin B12 in lung cancer, various population-based cohort studies measured circulating vitamin B12 concentrations in pre-diagnostic samples [246,247,248,249]. These studies found a positive correlation between cancer risk and the circulating concentration of vitamin B12 in cancer patients’ blood samples.

## 10. Conclusions and Future Perspectives

B vitamins are essential in maintaining cellular functions and metabolic processes. Their balance is very important for cell growth, proliferation, differentiation, and survival. Thus, understanding how B vitamins imbalance could lead to cancer growth and spread will help to control unnecessary dietary intake of excessive vitamin supplements. Further, identifying the role of vitamins is also crucial for developing chemopreventive strategies not only to maximize benefits but also to minimize potential risks. This review focused specifically on the relationship between B vitamins and cancer development or prevention. Several studies provided considerable evidence indicating both beneficial and harmful effects depending on the type of vitamin and cancer type [250,251,252]. While deficiencies in water-soluble B vitamins, such as thiamine, riboflavin, niacin, pantothenic acid, pyridoxine, biotin, folate, and cobalamin, are generally associated with an increased risk of cancer due to impaired metabolic and cellular functions, emerging research highlights the potential dangers related to excessive intake supplementation of these vitamins. Further, over-supplementation of B vitamins could enhance cancer cell metabolism, DNA repair, and genomic instability which can accelerate tumor progression in cancers with strong metabolic flexibility (Table 3).

For example, vitamins like folate and B12, which are essential for DNA synthesis and repair, and their excessive supplementation have been linked to increased risks of certain cancers, such as colorectal and lung cancer [247,248,249]. A recent case-control study by Le et al. [253] has suggested the risk of esophageal, lung, and breast cancers in patients with low intake of cobalamin and gastric cancer in patients with high intake in Vietnamese population. On the other hand, riboflavin and niacin have been shown to enhance the efficacy of cancer therapies [254,255]. Premkumar et al. [255] have shown that supplementation of riboflavin and niacin along with tamoxifen reduced tumor burden in breast cancer patients. Similarly, rapamycin and niacin combination increases apoptotic cell death in acute myeloid leukemia cells. [256]. Yuvaraj et al. [257] have also shown that combination treatment of Coenzyme-Q, riboflavin and niacin with tamoxifen e increases blood parameters in postmenopausal breast cancer women. However, the impact of vitamins B2 and B3 on cancer prevention and progression depends on dosage and cancer type.

Some risks exist in taking these B vitamins without proper guidance, as they can interact with certain medications and potentially alter their effectiveness. In addition to enhancing the chemotherapeutic effects of some drugs, some B vitamins could also improve the therapeutic effects of general medicines. For example, vitamin B12 supplementation can counteract the deficiency caused by long-term use of metformin [258]. Similarly, pyridoxine can reduce the efficacy of the anti-epileptic drug levodopa by increasing its metabolism [259]. Additionally, high doses of niacin have been shown to increase the risk of liver toxicity when taken along with cholesterol-lowering statins [260]. High niacin levels are also shown to be a risk for cardiovascular complications [261]. In addition, folic acid can interfere with methotrexate and reduce its effectiveness [262].

Further, recent studies also indicated several limitations and knowledge gaps that warrant attention on vitamin supplementation. A significant concern is the potential side effects of high-dose supplementation. A cohort study by Meyer et al. [263] has found that a combined high intake of pyridoxine and cobalamin was associated with an increased risk of hip fracture. Similarly, B vitamins might prevent cardiovascular diseases by lowering homocysteine levels; however, they can also alter the efficacy of statin drugs. Another limitation is the inconsistent results from clinical trials. While most observational studies have suggested the protective roles of certain B vitamins in preventing cancer growth, clinical trials have not consistently supported this, and additional studies are required. Further, there is a lack of comprehensive studies understanding the synergetic interactions between different B vitamins in human diseases. For example, taking folic acid alone could mask vitamin B12 deficiency, which could lead to neurological problems [264]. Thus, some of the discussed limitations point towards the need for more rigorous and well-designed studies to investigate the significance of B vitamins in human health and disease.

The interplay of B vitamins in cancer biology emphasizes the importance of maintaining an equilibrium in vitamin consumption. Careful attention is required to identify the cancer risk in patients with vitamin deficiency. Additional extensive population-based studies are needed to understand the dietary intake of vitamins and cancer progression and therapy. Understanding the role of vitamins in combination with chemotherapeutic drugs and immunotherapeutic drugs is required to enhance the therapeutic efficacy and increase the survival rate.

Further, identifying the molecular mechanisms of how the vitamins-mediated cellular and metabolic pathways are involved in cancer initiation and progression will help identify potential therapeutic strategies. Recent proteomic, metabolomic, and genomic studies will help identify novel biomarkers. Next-generation gene sequencing will also help to identify potential carcinogenic genes regulated by vitamin supplementation. Further studies are also needed to understand the interactions between different B vitamins and their combined effects on cancer progression, which could lead to more targeted and personalized approaches in cancer therapy.

Thus, future research should focus on elucidating the precise mechanisms by which these vitamins influence cancer pathways and identify safe and effective dose levels. Taking over-the-counter vitamin supplementation without a physician’s recommendation is sometimes a risk of developing unnecessary complications and side effects [265]. Since the requirement of vitamin supplementation depends on age, body metabolic rate, gender, and any prescribed medications, one should not self-diagnose and take supplements that unnecessarily can mask underlying health issues and complicate the disease progression [266]. Thus, physician input ensures vitamin supplements are safe, effective, and tailored to a patient’s needs. Therefore, always consult a healthcare provider before starting vitamin supplementation, who can recommend suggested doses based on specific medical conditions.

## Figures and Tables

**Figure 1 ijms-26-01967-f001:**
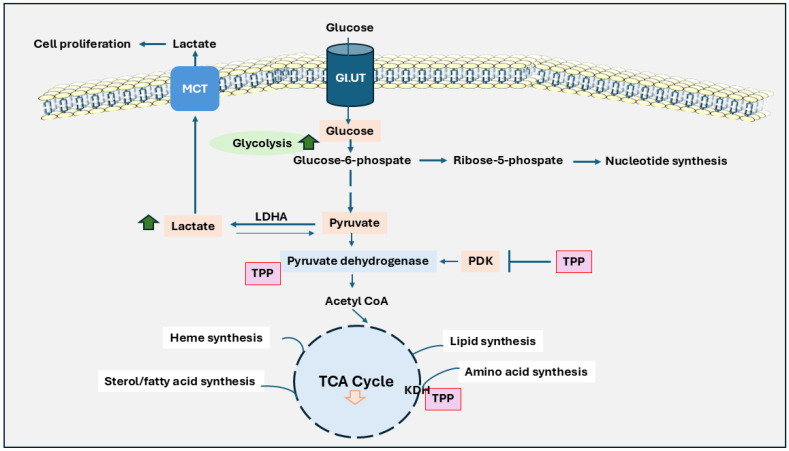
The role of thiamine in Warburg effect and its impact on cancer cell metabolism. Unlike normal cells, cancer cells heavily depend on glycolysis for energy production, even during aerobic conditions, and produce lactic acid instead of relying on oxidative phosphorylation. Glucose is taken up by the cell via GLUT transporters and converted through glycolysis into pyruvate. Instead of entering the TCA cycle within the mitochondria, pyruvate is converted to lactate in the cytoplasm by lactate dehydrogenase A (LDHA). Lactate is transported by monocarboxylate Transporters (MCT) and causes cancer cell proliferation. Further, the pyruvate dehydrogenase complex is inhibited by pyruvate dehydrogenase kinase (PDK), limiting pyruvate’s conversion to acetyl-CoA and reducing the flow into the TCA cycle. Thymine pyrophosphate (TPP) plays a critical role in restoring PDH activity and counteracting the inhibitory effects of PDK. This pathway also integrates alternative pathways, such as the pentose-phosphate pathway, hexosamine biosynthesis, and lipid synthesis, further supporting cancer cell growth and survival. The activity of the TCA cycle is generally reduced, which limits energy production from mitochondria. This metabolic adaptation provides rapid energy and intermediates needed for biosynthesis. As a result, cancer cells can grow and divide rapidly, even in challenging conditions like low oxygen levels.

**Figure 2 ijms-26-01967-f002:**
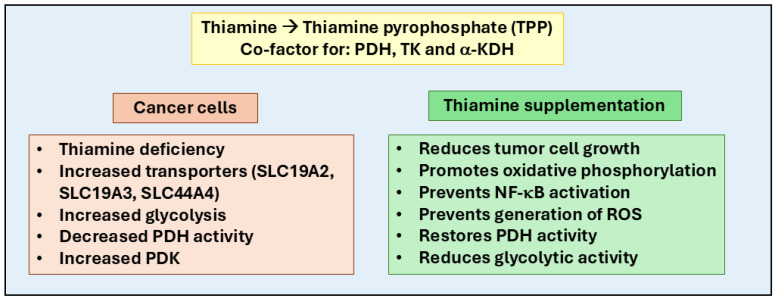
Role of thiamine in cancer progression and therapy. Thiamine (Vitamin B1) plays a crucial role in cellular metabolism by converting into thiamine pyrophosphate (TPP), a co-factor for enzymes like pyruvate dehydrogenase (PDH), transketolase (TK), and alpha-keto dehydrogenase (α-KDH). These enzymes regulate glycolysis, the pentose phosphate pathway, and the Krebs cycle, essential for ATP and NADPH production, DNA/RNA synthesis, and oxidative stress reduction. In cancer cells, thiamine influences the Warburg effect, where cancer cells rely on glycolysis even in oxygen-rich environments, often due to pyruvate dehydrogenase kinase (PDK) overexpression, which inhibits oxidative phosphorylation. Thiamine supplementation counters this by restoring PDH activity, reducing glycolytic dependence, and promoting oxidative metabolism. Cancer cells modulate thiamine uptake via transporters like SLC19A2, SLC19A3, and SLC44A4. Therapeutically, supplementation with thiamine and its derivatives (such as benfotiamine and oxythiamine) could inhibit tumor growth, reduce oxidative stress, prevent NF-κB-mediated inflammatory signaling, reduce glycolytic activity, and promote oxidative phosphorylation.

**Figure 3 ijms-26-01967-f003:**
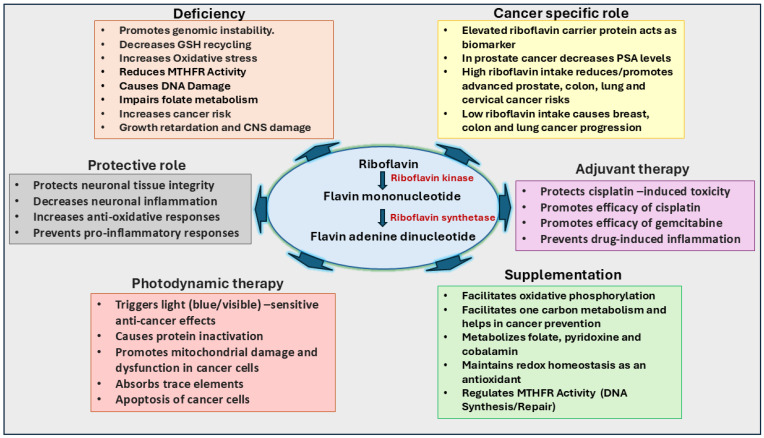
Significance of riboflavin in cancer growth and therapy. Riboflavin functions as a precursor for flavin mononucleotide (FMN) and flavin adenine dinucleotide (FAD), essential coenzymes in redox reactions, the TCA cycle, and reactive oxygen species (ROS) regulation. Riboflavin supports neural health, vitamin metabolism, and glutathione maintenance. Its deficiency, termed ariboflavinosis, can increase oxidative stress and promote cancer risk by impairing folate metabolism and DNA synthesis. In cancer therapy, riboflavin regulates ROS to inhibit tumor growth, promotes apoptosis, and reduces inflammation, enhancing the efficacy of cisplatin and gemcitabine. Photosensitive properties of riboflavin increase its anti-cancer potential. Elevated riboflavin intake correlates with reduced risks for colorectal, lung, and cervical cancers, while high serum riboflavin levels may increase risks for pancreatic and colorectal cancers.

**Figure 4 ijms-26-01967-f004:**
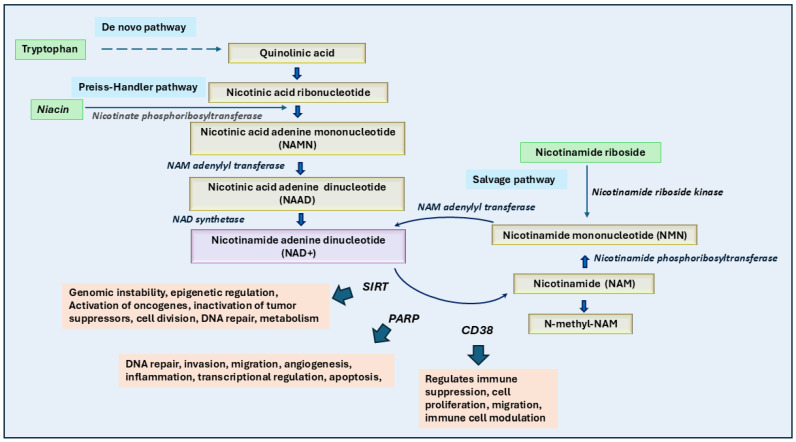
Role of niacin and its cofactors in cancer growth and therapy. The biosynthesis and metabolism of NAD+ (nicotinamide adenine dinucleotide) occur through three main pathways: the de novo pathway (from tryptophan to quinolinic acid and eventually NAD+), the Preiss-Handler pathway (from niacin to NAD+ via intermediates like NAMN and NAAD), and the salvage pathway (from nicotinamide riboside or nicotinamide to NAD+ via NMN). Enzymatic reactions, such as those catalyzed by nicotinamide riboside kinase and nicotinate phosphoribosyltransferase, play critical roles in these conversions. NAD+ serves as a cofactor for key enzymes like sirtuins (SIRTs), Poly (ADP-ribose) polymerases (PARPs), and CD38, which are involved in genomic stability, DNA repair, cell proliferation, and immune suppression, which eventually lead to cancer growth. NAMPT inhibitors, when combined with niacin, effectively block NAD salvage pathways crucial for tumor growth. High dietary niacin intake could correlate with improved survival and lower mortality in cancer patients, supporting its potential as a chemopreventive agent.

**Figure 5 ijms-26-01967-f005:**
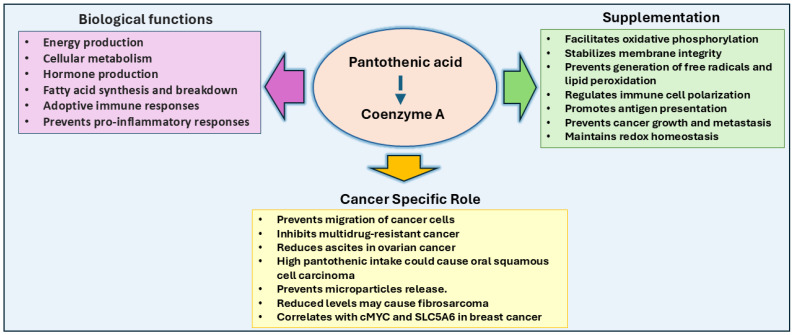
Pantothenic acid in cancer: Pantothenic acid is a precursor to Coenzyme A (CoA) and is vital for numerous metabolic processes, including the citric acid cycle, fatty acid metabolism, and hormone production. Increased pantothenic acid levels correlate with enhanced glycolytic activity and cancer cell migration, particularly in breast and gastric cancer cells, and its association with c-MYC and SLC5A6 expression promotes breast cancer growth. Pantothenic acid supplementation could enhance immune responses by increasing antigen presentation, shifting tumor metabolism from glycolysis to oxidative phosphorylation, and preventing tumor growth. High intake has been linked to oral squamous cell carcinoma risk, and pantothenate supplementation has been shown to treat multidrug-resistant cancers and reduce metastasis.

**Figure 6 ijms-26-01967-f006:**
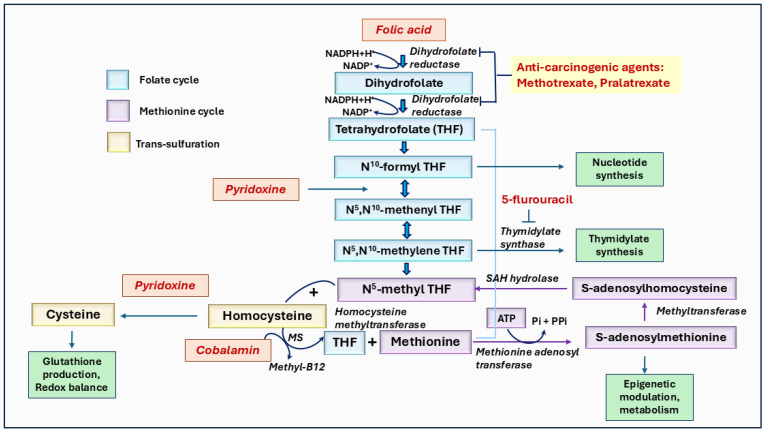
The role of vitamins B6, B9, and B12 mediated one-carbon metabolism in cancer. Folate (vitamin B9) is converted to dihydrofolate and then to tetrahydrofolate (THF), which undergoes further modifications to form derivatives like N^5^, N^10^-methylene THF, and N^5^-methyl THF. These derivatives are essential for methylation processes, and nucleotide synthesis requires cancer cell growth. The pathway highlights the involvement of pyridoxine (vitamin B6) in converting homocysteine to cysteine via the trans-sulfuration pathway and cobalamin (vitamin B12) in the re-methylation of homocysteine to methionine, which subsequently forms S-adenosylmethionine (SAM), a key methyl donor in several biochemical and epigenetic methylations. Anti-carcinogenic agents, such as methotrexate and pralatrexate, inhibit steps in folate metabolism, while 5-fluorouracil blocks nucleotide synthesis. This pathway underscores the interdependence of folate, pyridoxine, and cobalamin in one-carbon metabolism in essential biological functions like methylation, DNA synthesis, and cellular regulation.

**Table 1 ijms-26-01967-t001:** Role of B vitamins in various metabolic and cancer-related pathways.

Vitamin	Key Role	Type of Co-Factor Formed	Key Step in Metabolism	Role in Cancer-Related Pathways
B1 (Thiamine)	Decarboxylation of α-keto acids, transketolase reactions	Thiamine pyrophosphate (TPP)	Pyruvate dehydrogenase complex: Pyruvate → Acetyl-CoA	Thiamine deficiency leads to impaired mitochondrial function and altered glucose metabolism which could promote cancer cell survival.
B2 (Riboflavin)	Redox reactions (electron transfer)	Flavin adenine dinucleotide (FAD), Flavin mononucleotide (FMN)	TCA cycle: Succinate dehydrogenase at step Succinate → Fumarate	Riboflavin deficiency could enhance oxidative DNA damage and increase cancer risk.
B3 (Niacin)	Redox reactions (electron transfer)	Nicotinamide adenine dinucleotide (NAD), Nicotinamide adenine dinucleotide phosphate (NADP)	Glycolysis: Glyceraldehyde-3-phosphate dehydrogenase step (Glyceraldehyde-3-phosphate → 1,3-Bisphosphoglycerate)	Niacin deficiency could impair DNA repair and increase mutation rates.
B5 (Pantothenic Acid)	Acyl group transfer	Coenzyme A (CoA)	Fatty acid synthesis: Formation of Malonyl-CoA from Acetyl-CoA	Altered CoA in cancer cells promote proliferation and membrane biosynthesis.
B6 (Pyridoxine)	Amino acid metabolism (transamination, decarboxylation)	Pyridoxal phosphate (PLP)	Transamination: Conversion of Aspartate to Oxaloacetate	PLP deficiency can increase homocysteine levels and impair DNA synthesis, which contribute to tumorigenesis.
B7 (Biotin)	Carboxylation reactions (gluconeogenesis, fatty acid synthesis)	Biotin-enzymes complex	Gluconeogenesis: Pyruvate carboxylase reaction (Pyruvate → Oxaloacetate)	Biotin is involved in histone modification which affects gene regulation and cancer cell growth.
B9 (Folate)	One-carbon metabolism (DNA synthesis)	Tetrahydrofolate (THF)	DNA synthesis: Methylation of deoxyuridylate to form thymidylate (dUMP → dTMP)	Folate deficiency can cause DNA strand breaks and an aberrant gene expression which increase cancer risk.
B12 (Cobalamin)	Methylation and rearrangement reactions	Methylcobalamin, 5′-deoxyadenosylcobalamin	Methionine synthesis: Conversion of homocysteine to methionine	B12 deficiency can lead to genome instability and increase cancer susceptibility.

**Table 2 ijms-26-01967-t002:** Role of B vitamins in Warburg effect. Vitamins B6, B9, and B12 are not directly involved in the Warburg effect. They are directly involved in amino acid metabolism, one-carbon metabolism, and nucleic acid synthesis.

B Vitamin	Key Enzymes & Intermediates	Impact on the Warburg Effect	Interaction with Oncogenic Metabolic Regulators
B1	Pyruvate dehydrogenase (PDH) Transketolase (TKT)	Deficiency: Inhibits PDH, increase pyruvate-to-lactate conversion, reinforce glycolysis. Excess: Enhances PDH activity, promoting oxidative phosphorylation (OXPHOS) and reducing glycolysis.	HIF-1α: Upregulates PDK, which inhibits PDH, favor glycolysis over oxidative phosphorylation. MYC: Activates TKT, promotes pentose phosphate pathway, increases nucleotide synthesis
B2	Succinate dehydrogenase (SDH, ETC Complex II) NADH dehydrogenase (ETC Complex I)	Deficiency: Impairs ETC function, increase glycolysis. Excess: Supports mitochondrial function, promote oxidative phosphorylation.	AMPK: riboflavin deficiency leads to AMPK activation and mitochondrial dysfunction, promotes glycolysis.
B3	NAD+/NADH balance Sirtuins (SIRT1, SIRT3) PARPs	Deficiency: Reduces NAD+/NADH ratio, impairs TCA cycle, promotes glycolysis. Excess: Increases NAD+, enhances mitochondrial function, reverses the Warburg effect.	HIF-1α: NAD+ depletion stabilizes HIF-1α, enhancing glycolysis. SIRT1: Inhibits HIF-1α and promotes oxidative phosphorylation. PARP1: Initiates DNA repair mechanisms, maintains genomic stability.
B5	Acetyl-CoA synthase Fatty acid synthase (FASN)	Deficiency: Reduces acetyl-CoA availability, increases glycolysis dependence. Excess: Enhances lipid metabolism, supports tumor growth	MYC: Increases FASN expression, promotes lipid metabolism for cell proliferation. AMPK: Low acetyl-CoA triggers AMPK activation, promotes glycolysis.
B6	Glutaminase (GLS) Serine hydroxy-methyltransferase (SHMT)	Deficiency: Impairs glutamine metabolism in TCA cycle, reinforces glycolysis. Excess: Enhances one-carbon metabolism, supports oxidative phosphorylation.	MYC: Upregulates GLS, increases glutamine metabolism, promotes cell proliferation.
B7	Pyruvate carboxylase (PC)	Deficiency: Reduces oxaloacetate, impairs TCA cycles, promotes glycolysis. Excess: Promotes mitochondrial metabolism and TCA cycle.	HIF-1α: Suppress PC expression, diverts pyruvate to lactate.
B9	Thymidylate synthase (TS)—Methylenetetrahydrofolate reductase (MTHFR)	Deficiency: Impairs nucleotide synthesis, causes DNA damage and metabolic stress. Excess: Promotes nucleotide biosynthesis, increases tumor growth.	MYC: Enhances folate metabolism, increases cell proliferation. HIF-1α: Induces nucleotide synthesis genes, increases folate demand.
B12	Methionine synthase (MS)	Deficiency: Disrupts methylation, alters metabolic enzyme expression. Excess: Supports epigenetic regulation, promotes tumor survival.	MYC: Upregulates methionine cycle enzymes, promotes epigenetic modifications and oncogene expressions.

**Table 3 ijms-26-01967-t003:** Effects of B Vitamin Deficiencies and Over-Supplementation on Cancer Progression and Prevention.

B Vitamin	Effects of Deficiency	Effects of Over-Supplementation
B1 (Thiamine)	Impairs mitochondrial function, decreases oxidative phosphorylation, increases oxidative stress and metabolic dysfunction, inhibits tumor growth [32,33,34,35,42,45,46,47,48,49].	Enhances cancer cell metabolism by promoting glycolysis and the pentose phosphate pathway and accelerates cancer progression in specific cancers [50,51,52,67].
B2 (Riboflavin)	Reduces flavoprotein activity, increases oxidative stress and DNA damage; impairs Krebs cycle; inhibits tumor progression in some cancers [70,71,72,73,86,87,88,89].	Enhances Krebs cycle and oxidative phosphorylation, reduces free radical production, promotes cancer cell growth, and inhibits oxidative stress-induced apoptosis [74,75,76,77,78,79,80,81,93,94].
B3 (Niacin)	Reduces NAD+ levels, impairs DNA repair, inhibits PARP and SIRT activation, increases genomic instability and promotes tumorigenesis [113,114,115,116,129].	Increases NAD+ levels, enhances cancer metabolism and DNA repair; may prevent cancer progression in some cases but also promote tumor survival in therapy-resistant cancers [123,124,125,126,144].
B5 (Pantothenic Acid)	Disrupts Coenzyme A (CoA) production, impairs fatty acid synthesis and energy metabolism, and prevents tumor growth [147,148,149,155].	Enhances lipid metabolism, promotes cancer cell proliferation and progression [150,151,152,153,154,155,156,160].
B6 (Pyridoxine)	Reduces amino acid metabolism and neurotransmitter synthesis; increases homocysteine levels which lead to inflammatory responses that may promote cancer progression [165,166,167,168,177,178,179,180].	Stimulates angiogenesis and increases cancer cell survival in specific conditions [169,170,171,172,173,174,175,176,182,183,184,185,186,187].
B7 (Biotin)	Decreases biotin-dependent carboxylase activities, limits fatty acid synthesis and prevents tumor growth [188,189,193,194].	Promotes biotinylation, affects oncogene expression, and contributes to tumor growth [196,197,198,199].
B9 (Folate)	Impairs one-carbon metabolism, hinders DNA synthesis and repair, increases mutation rates and cancer risk [201,202,203,204,207,208,209,210,211,212].	Enhances one-carbon metabolism, promotes tumor progression by providing nucleotides for DNA replication [219,220,221,222,223,224,225,226,227,228,229,230].
B12 (Cobalamin)	Causes genomic instability due to impaired methylation and DNA synthesis, increases cancer risk and progression [231,232,233,234,239].	Supports cancer cell metabolism and promote the risk of certain cancers [235,236,237,238,239,240,241,242,243,244,245,246].

## Data Availability

Not applicable.

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
