# Peer review of "Recent Advances on the Role of B Vitamins in Cancer Prevention and Progression"

_ijms, 2025, doi:10.3390/ijms26051967_

Round 1

Reviewer 1 Report

Comments and Suggestions for Authors

Journal: IJMS (ISSN 1422-0067)

Manuscript ID: ijms-3480298

Type: Review

Title:Water-Soluble B Vitamins in Cancer Prevention and Progression: Benefits and Risks

The manuscript titled "Water-Soluble B Vitamins in Cancer Prevention and Progression: Benefits and Risks" by Zachary Frost et al. provides a well-organized, thorough, and informative analysis of the dual role of B vitamins in cancer biology. The authors expertly combine a wide range of study findings, offering a fair discussion of the potential advantages and hazards of B vitamin intake for cancer prevention and progression.

Introduction:

1: Separate discussions about how B vitamins help with cancer prevention and progression.

2: Extend the risks associated with a high intake of all B vitamins, not simply folate and B12.

3: Insert a column in the table that links the B vitamin's role to cancer-related pathways.

2. Vitamin B1: Thiamine

4: The section focuses on thiamine's favorable metabolic effects (ATP production, ribose synthesis, and NADPH synthesis), but it does not discuss the possibility that thiamine may accelerate cancer progression in certain cases.

Consider that, depending on the type of cancer and metabolic state, thiamine can have both tumor-promoting and tumor-suppressive effects.

Figure 1: How does TPP supplementation affect PDH activity in cancer cells?

5: Lines 195 to 198: The mention of FAD/FMN modulation, ROS control, and apoptosis induction indicates that the mechanisms are at least somewhat understood. The language is deceptive since it minimizes existing knowledge while citing numerous studies that support riboflavin's impact on cancer.

Reframe this statement to reflect that, while some processes have been established, further research is needed to understand riboflavin's dose-dependent and cancer-specific effects.

5. Vitamin B5: Pantothenic Acid

6: Initially, the article claims that pantothenic acid has no substantial impact on cancer risk, citing research on esophageal, gastric, urothelial, and breast cancers (lines 378-389). However, later portions imply that pantothenic acid stimulates cancer cell proliferation, notably in breast and stomach malignancies (lines 391-412).

Clearly describe whether pantothenic acid is neutral, useful, or detrimental. Consider breaking into different subsections for clarification.
"No significant association with cancer risk"
"Potential cancer-promoting effects" 
"Therapeutic potential in cancer treatment"

7:   While some studies (e.g., Lines 401-412) report results from cell lines and murine models, subsequent sections (Lines 414-427) address clinical correlations with anti-PD1 therapy without clearly identifying which results are observational and which are test results for therapeutic interventions in humans.
Indicate clearly which studies have been evaluated in human clinical settings and which are preclinical (cell culture/animal studies). This differentiation will assist readers in determining the findings' pertinence.

8. Vitamin B9: Folate

8: Some claims, like "Folate has been shown to be critical in regulating the cancer growth," might be strengthened by adding more citations. Would you think about including more detailed references in this section?

Conclusions and Future Perspectives:

9: Niacin and riboflavin are said to "enhance the efficacy of cancer therapies" in the text, but no circumstances, processes, or settings are mentioned.
 While some B vitamins may improve specific treatments in some situations, they may also conflict with other drugs. The way these vitamins work with various treatments has to be clarified further.

Author Response

The manuscript titled "Water-Soluble B Vitamins in Cancer Prevention and Progression: Benefits and Risks" by Zachary Frost et al. provides a well-organized, thorough, and informative analysis of the dual role of B vitamins in cancer biology. The authors expertly combine a wide range of study findings, offering a fair discussion of the potential advantages and hazards of B vitamin intake for cancer prevention and progression.

 Introduction:

1: Separate discussions about how B vitamins help with cancer prevention and progression.

    As suggested, we have included a separate paragraph in the introduction

2: Extend the risks associated with a high intake of all B vitamins, not simply folate and B12.

  As suggested, we have revised the sentence.

3: Insert a column in the table that links the B vitamin's role to cancer-related pathways.

As suggested, we have included an additional column in the revised table showing cancer-related pathways. The table has been included at the end of the references section.

  1. Vitamin B1: Thiamine

4: The section focuses on thiamine's favorable metabolic effects (ATP production, ribose synthesis, and NADPH synthesis), but it does not discuss the possibility that thiamine may accelerate cancer progression in certain cases. Consider that, depending on the type of cancer and metabolic state, thiamine can have both tumor-promoting and tumor-suppressive effects.

As suggested, we have discussed the role of thiamine in both tumor promoting and suppressing effects. The following information has been added to the section.

Thus, depending on the type of cancer and metabolic state, thiamine can have both tumor-promoting and tumor-suppressive effects. For example, thiamine supports tumor growth by enhancing glycolysis, the pentose phosphate pathway, and mitochondrial function and providing ATP, NADPH, and biosynthetic precursors essential for rapid tumor cell proliferation. However, in some conditions, thiamine deficiency could cause metabolic stress and oxidative DNA damage, potentially suppressing tumor growth. Although adequate dietary thiamine is necessary for normal function, excessive supplementation could promote tumor progression in susceptible cancers, and personalized approaches are needed when using thiamine in cancer therapy.

Figure 1: How does TPP supplementation affect PDH activity in cancer cells?

TPP supplementation enhances PDH activity in cancer cells primarily when thiamine levels are deficient.  Depending on the metabolic flexibility of the tumor, this can either suppress tumor growth by reducing glycolysis or promote growth by enhancing mitochondrial metabolism.

5: Lines 195 to 198: The mention of FAD/FMN modulation, ROS control, and apoptosis induction indicates that the mechanisms are at least somewhat understood. The language is deceptive since it minimizes existing knowledge while citing numerous studies that support riboflavin's impact on cancer.Reframe this statement to reflect that, while some processes have been established, further research is needed to understand riboflavin's dose-dependent and cancer-specific effects.

As suggested, we have revised the statements.

  1. Vitamin B5: Pantothenic Acid

6: Initially, the article claims that pantothenic acid has no substantial impact on cancer risk, citing research on esophageal, gastric, urothelial, and breast cancers (lines 378-389). However, later portions imply that pantothenic acid stimulates cancer cell proliferation, notably in breast and stomach malignancies (lines 391-412). Clearly describe whether pantothenic acid is neutral, useful, or detrimental. Consider breaking into different subsections for clarification.
"No significant association with cancer risk"
"Potential cancer-promoting effects" 
"Therapeutic potential in cancer treatment"

 As suggested, we have carefully revised this section.

7:   While some studies (e.g., Lines 401-412) report results from cell lines and murine models, subsequent sections (Lines 414-427) address clinical correlations with anti-PD1 therapy without clearly identifying which results are observational and which are test results for therapeutic interventions in humans.  Indicate clearly which studies have been evaluated in human clinical settings and which are preclinical (cell culture/animal studies). This differentiation will assist readers in determining the findings' pertinence.

As suggested, we have indicated pre-clinical and clinical studies.

  1. Vitamin B9: Folate

8: Some claims, like "Folate has been shown to be critical in regulating the cancer growth," might be strengthened by adding more citations. Would you think about including more detailed references in this section?

 As suggested, we have included additional references to the statement.  

Conclusions and Future Perspectives:

9: Niacin and riboflavin are said to "enhance the efficacy of cancer therapies" in the text, but no circumstances, processes, or settings are mentioned.
While some B vitamins may improve specific treatments in some situations, they may also conflict with other drugs. The way these vitamins work with various treatments has to be clarified further.

As suggested, we have carefully rewritten our statements.

Reviewer 2 Report

Comments and Suggestions for Authors

The manuscript by Frost et al. addresses a pertinent topic: the dual roles of water-soluble B vitamins in cancer prevention and progression. This subject is of significant interest because B vitamins function as coenzymes or precursors in various metabolic processes, including energy production, nerve function, and DNA synthesis. However, the manuscript would benefit from a deeper mechanistic perspective on how individual B vitamins influence cancer development and treatment.

Specific Comments:

  1. The manuscript provides a general overview of the roles of B vitamins in metabolic processes and their association with cancer. To deepen the discussion, the authors should explore specific mechanisms by which individual B vitamins affect cancer biology. For instance, they should discuss the role of Folate (B9) in one-carbon metabolism, nucleotide biosynthesis, and DNA methylation, and how these processes influence tumorigenesis. Additionally, they should elaborate on the involvement of B6 in amino acid metabolism and its potential impact on cancer cell proliferation and apoptosis. Regarding B12, explore its function in DNA synthesis and methylation, and the implications of its deficiency or excess in cancer risk.
  2. Examine the role of Niacin (B3) in the formation of NAD+ and NADH, crucial for cellular metabolism and DNA repair mechanisms, and how this relates to cancer progression. NAD+/NADH balance, is critical in the context of the Warburg effect and cancer cell metabolism.
  3. Abstract lines 14-15 and elsewhere within the text: The manuscript does not clearly distinguish between the effects of B vitamin deficiencies and over-supplementation. Each state can have drastically different implications for cancer risk and progression, and a clearer distinction could aid in understanding the dual roles these vitamins play in both promoting and inhibiting cancer. It would be beneficial to distinguish between the effects of B vitamin deficiencies versus over-supplementation on tumor development, as both scenarios can have contrasting outcomes.
  4. Lines 74-91, Figure 1: The connection between the Warburg effect and B vitamins is not clearly articulated, and Figure 1 does not effectively illustrate this relationship. The authors should provide a detailed explanation of how specific B vitamins influence the Warburg effect. For example, discuss how Thiamine (B1) affects pyruvate dehydrogenase activity, thereby modulating the shift from glycolysis to oxidative phosphorylation.
  5. In light of the above comment, the authors should update Figure 1 to include key enzymes and intermediates involved in metabolic pathways affected by B vitamins. Highlight interactions between B vitamins and oncogenic metabolic regulators such as MYC, HIF-1α, and AMPK.
  6. Make a clear distinction between epidemiological findings and mechanistic studies to prevent mixing up observational data with conclusions about cause and effect. 
  7. Expand the discussion on how B vitamin metabolism may influence chemotherapy resistance; e.g. discuss how alterations in folate metabolism can affect the efficacy of antifolate chemotherapeutic agents.
  8. Please use more recent studies to provide up-to-date evidence supporting the manuscript's claims, particularly regarding the complex roles of B vitamins in cancer prevention and progression.

Author Response

Reviewer 2:

The manuscript by Frost et al. addresses a pertinent topic: the dual roles of water-soluble B vitamins in cancer prevention and progression. This subject is of significant interest because B vitamins function as coenzymes or precursors in various metabolic processes, including energy production, nerve function, and DNA synthesis. However, the manuscript would benefit from a deeper mechanistic perspective on how individual B vitamins influence cancer development and treatment.

Specific Comments:

  1. The manuscript provides a general overview of the roles of B vitamins in metabolic processes and their association with cancer. To deepen the discussion, the authors should explore specific mechanisms by which individual B vitamins affect cancer biology. For instance, they should discuss the role of Folate (B9) in one-carbon metabolism, nucleotide biosynthesis, and DNA methylation, and how these processes influence tumorigenesis. Additionally, they should elaborate on the involvement of B6 in amino acid metabolism and its potential impact on cancer cell proliferation and apoptosis. Regarding B12, explore its function in DNA synthesis and methylation, and the implications of its deficiency or excess in cancer risk.

As suggested, we have expanded this information in the respective sections (please see the highlighted portions)

  1. Examine the role of Niacin (B3) in the formation of NAD+ and NADH, crucial for cellular metabolism and DNA repair mechanisms, and how this relates to cancer progression. NAD+/NADH balance, is critical in the context of the Warburg effect and cancer cell metabolism.

In cancer cells, the Warburg effect shifts glucose metabolism towards glycolysis, which alters the NAD+/NADH ratio to promote cell proliferation. Further, NAD+ is also crucial for PARP enzyme activity involved in DNA repair and cancer progression. Thus, the therapies targeting NAD+ metabolism, such as PARP and NAMPT inhibitors, could enhance cancer cell death by blocking NAD+ dependency. Thus, the modulation of NAD+/NADH balance demonstrates a promising therapeutic strategy, but its effects depend on the specific cancer type and treatment approach. This information has been included in the B3 section (please see the highlighted portion).

  1. Abstract lines 14-15 and elsewhere within the text: The manuscript does not clearly distinguish between the effects of B vitamin deficiencies and over-supplementation. Each state can have drastically different implications for cancer risk and progression, and a clearer distinction could aid in understanding the dual roles these vitamins play in both promoting and inhibiting cancer. It would be beneficial to distinguish between the effects of B vitamin deficiencies versus over-supplementation on tumor development, as both scenarios can have contrasting outcomes.

As suggested, we have included a table distinguishing between the effects of B vitamin deficiencies versus over-supplementation on tumor development in the conclusions section. (please see Table 3)

  1. Lines 74-91, Figure 1: The connection between the Warburg effect and B vitamins is not clearly articulated, and Figure 1 does not effectively illustrate this relationship. The authors should provide a detailed explanation of how specific B vitamins influence the Warburg effect. For example, discuss how Thiamine (B1) affects pyruvate dehydrogenase activity, thereby modulating the shift from glycolysis to oxidative phosphorylation.

Figure -1 discusses the role of TPP in the Warburg effect. However, as suggested, we have included an additional table showing the significance of all the B vitamins in the Warburg effect.  (please see Table 2)

  1. In light of the above comment, the authors should update Figure 1 to include key enzymes and intermediates involved in metabolic pathways affected by B vitamins. Highlight interactions between B vitamins and oncogenic metabolic regulators such as MYC, HIF-1α, and AMPK.

Inclusion of all the enzymes and oncogenic regulators in Figure-1 complicate the role of Thiamine alone depicted in the figure. Instead, as suggested, we have provided a new table (Table-2) describing comments 4 and 5

  1. Make a clear distinction between epidemiological findings and mechanistic studies to prevent mixing up observational data with conclusions about cause and effect. 

As suggested, we have avoided mixing up with epidemiological and observational studies.

  1. Expand the discussion on how B vitamin metabolism may influence chemotherapy resistance; e.g. discuss how alterations in folate metabolism can affect the efficacy of antifolate chemotherapeutic agents.

As suggested, we have expanded discussion on how B vitamin metabolism may influence chemotherapy resistance

  1. Please use more recent studies to provide up-to-date evidence supporting the manuscript's claims, particularly regarding the complex roles of B vitamins in cancer prevention and progression.

As suggested, we have updated with more recent citations (highlighted the recent citations)

Reviewer 3 Report

Comments and Suggestions for Authors

1. The title should be improved to reflect advances in the role of B vitamins in cancer prevention and progression.
2. In lines 7-19 of the abstract, key findings of the review that may enhance its informative impact can be included. The main objective and the approach used should also be included. 
3. You should use keywords other than those in the title and increase the number of keywords used.
4. The introduction should improve the connection between paragraphs for a smoother reading. The importance of B vitamins in cancer prevention should also be more explicitly connected. Likewise, the review's objectives should be better delimited, and its unique contribution to other reviews should be highlighted.
5. Table 1 on line 33 is unnecessary. The information should be written in a paragraph, as it is not common to put tables in an introduction to a review article.
6. MDPI usually uses the PRISMA methodology to write review articles. It is suggested that the methods used in the review be described, including criteria for study selection, databases consulted, and analytical approaches used (this paragraph can be placed in part of the introduction).
7. It is suggested that the graphic representation of the figures be improved by using professional tools such as BIORENDER or similar. There is too much text in the representation.
8. The relationship between deficiency of B vitamins and increased risk of cancer should be justified in greater depth. 
9. More details on the side effects of excessive intake of specific B vitamins should also be included, as well as clarification on whether the effects of B vitamins depend on the patient's genetic and metabolic content.
10. In the section on B-vitamin metabolism and mechanisms of action, the limitations of previous studies and identified knowledge gaps should be discussed, as well as the bioavailability of B vitamins from different dietary sources.
11. Summary tables with easy-to-understand information and with authors cited are also typically used in review articles.
12. Regarding therapeutic applications and clinical considerations, the selected studies should be evaluated to determine whether they justify using B vitamin supplements in oncologic patients.
13. The discussion on the relationship between B-vitamin metabolism and resistance to oncological therapies should be expanded.
14. Conclusions should synthesize the study's key findings. They should be connected with the objectives stated in the introduction, and future lines of research on the relationship between B vitamins and cancer should be specifically proposed.
15. It is recommended that an explicit section on the study's limitations be included (it can be in the conclusions).
16. Improve the cohesion between the article sections and evaluate if the structure presented is correct.
17. The ITHENTICATE similarity index is high. Evaluate the possibility of paraphrasing.

Author Response

  1. The title should be improved to reflect advances in the role of B vitamins in cancer prevention and progression.

As suggested, we have changed the title to “Recent Advances on the role of B Vitamins in Cancer Prevention and Progression.”

  1. In lines 7-19 of the abstract, key findings of the review that may enhance its informative impact can be included. The main objective and the approach used should also be included. 

As suggested, we have revised the abstract

  1. You should use keywords other than those in the title and increase the number of keywords used.

As suggested, we have included additional keywords

  1. The introduction should improve the connection between paragraphs for a smoother reading. The importance of B vitamins in cancer prevention should also be more explicitly connected. Likewise, the review's objectives should be better delimited, and its unique contribution to other reviews should be highlighted.

As suggested, we have carefully revised introduction for smooth reading and included importance of B vitamins in cancer prevention.

  1. Table 1 on line 33 is unnecessary. The information should be written in a paragraph, as it is not common to put tables in an introduction to a review article.

As suggested, we have rewarded the table title.

  1. MDPI usually uses the PRISMA methodology to write review articles. It is suggested that the methods used in the review be described, including criteria for study selection, databases consulted, and analytical approaches used (this paragraph can be placed in part of the introduction).

As suggested, this information has been included (please see the highlighted section in the Introduction).

  1. It is suggested that the graphic representation of the figures be improved by using professional tools such as BIORENDER or similar. There is too much text in the representation.

This is a nice suggestion; we would consider this suggestion for future articles.  

  1. The relationship between deficiency of B vitamins and increased risk of cancer should be justified in greater depth. 

Please see the new table added to the revised manuscript with the details.

  1. More details on the side effects of excessive intake of specific B vitamins should also be included, as well as clarification on whether the effects of B vitamins depend on the patient's genetic and metabolic content.

As suggested, we have included this information in conclusion sections

  1. In the section on B-vitamin metabolism and mechanisms of action, the limitations of previous studies and identified knowledge gaps should be discussed, as well as the bioavailability of B vitamins from different dietary sources.

As suggested, we have included this information in the introduction section and Table-1, and introductory sections on each vitamin.

  1. Summary tables with easy-to-understand information and with authors cited are also typically used in review articles.

As suggested, we have included two additional tables.

  1. Regarding therapeutic applications and clinical considerations, the selected studies should be evaluated to determine whether they justify using B vitamin supplements in oncologic patients.

This has been discussed in the conclusion section.

  1. The discussion on the relationship between B-vitamin metabolism and resistance to oncological therapies should be expanded.

As suggested, we have expanded the information in each section.

  1. Conclusions should synthesize the study's key findings. They should be connected with the objectives stated in the introduction, and future lines of research on the relationship between B vitamins and cancer should be specifically proposed.

As suggested, we have revised the conclusions section.

  1. It is recommended that an explicit section on the study's limitations be included (it can be in the conclusions).

As suggested, we have included this information in the conclusion section.

  1. Improve the cohesion between the article sections and evaluate if the structure presented is correct.

We have carefully selected the sections and discussed the role of each b vitamin

  1. The ITHENTICATE similarity index is high. Evaluate the possibility of paraphrasing.

As suggested, we have carefully edited the manuscript.

Round 2

Reviewer 2 Report

Comments and Suggestions for Authors

The authors have addressed all my queries, and the manuscript has been significantly improved following the first round of revisions. I have no further comments, and in my opinion, the manuscript can be accepted for publication in its current form.

Author Response

The authors have addressed all my queries, and the manuscript has been significantly improved following the first round of revisions. I have no further comments, and in my opinion, the manuscript can be accepted for publication in its current form.

Thank you for finding our manuscript improved after revisions and acceptable for publication. 

Reviewer 3 Report

Comments and Suggestions for Authors

Dear Authors,
I appreciate your efforts in improving the manuscript and consider that you have addressed the comments raised in general.
However, I would like to insist on improving the figures before publication. Visual representation of concepts is critical in review articles, and graphical editing tools can help present information more clearly and professionally.
I understand that you mentioned that you will consider it for future studies. Still, if you could make some adjustments in this version, I think it would significantly strengthen the article's visual impact.

Author Response

I appreciate your efforts in improving the manuscript and consider that you have addressed the comments raised in general.
However, I would like to insist on improving the figures before publication. Visual representation of concepts is critical in review articles, and graphical editing tools can help present information more clearly and professionally.
I understand that you mentioned that you will consider it for future studies. Still, if you could make some adjustments in this version, I think it would significantly strengthen the article's visual impact.

As suggested, we have improved the clarity of the images. Since all the figures were submitted in the PDF format, their resolution might be reduced. Therefore, we have also submitted the figures in better resolution TIFF format (please see supplement files)